# Quality Traits and Nutritional Components of Cherry Tomato in Relation to the Harvesting Period, Storage Duration and Fruit Position in the Truss

**DOI:** 10.3390/plants12020315

**Published:** 2023-01-09

**Authors:** Pavlos Tsouvaltzis, Stela Gkountina, Anastasios S. Siomos

**Affiliations:** 1Department of Horticulture, Aristotle University, 54124 Thessaloniki, Greece; 2New South Wales Department of Primary Industries, Ourimbah, NSW 2258, Australia

**Keywords:** vegetables, color, firmness, antioxidants, carotenoids, lycopene, phenols

## Abstract

It is well known that the harvesting period and the storage duration have a significant effect on the quality characteristics of cherry tomato fruits. On the other hand, the effect of the fruit position in the truss has not been studied, as well as the relative contribution of each one of these factors on fruit quality. For this purpose, cherry tomato (Genio F1) whole trusses were harvested at the fruit red ripe stage during three periods. At each harvesting period, the first four (at the base of the truss) and the last four (at the top) fruits from each truss that was previously trimmed to 10 fruits, were stored at 12 °C for 0, 4 and 10 days. At the end of each storage duration, the external color, firmness, antioxidant capacity, pH and titratable acidity, as well as dry matter, soluble solid, total soluble phenol, lycopene, total carotenoid and β-carotene content, were determined. Analysis of variance (ANOVA) indicated that the harvesting period had the most significant effect on skin color parameters L * and C * and β-carotene, as well as on antioxidant capacity, total soluble phenols, dry matter and total soluble solids, while it also had an appreciable effect on titratable acidity. The storage duration had a dominant effect on firmness, total carotenoids and lycopene, while it had an appreciable effect on skin color parameter L * as well. On the other hand, the fruit position in the truss exerted an exclusive effect on h^o^ and a */b * ratio skin color parameters and pH and an appreciable effect on titratable acidity.

## 1. Introduction

The tomato (*Solanum lycopersicum* L.) belongs to the Solanaceae family and is one of the most important vegetables both in terms of consumption and nutritional value. Several epidemiological studies have proven that tomato consumption reduces the risk of some chronic diseases [1,2]. The nutritional value of the tomato is attributed to components of high antioxidant capacity, such as lycopene, β-carotene, total soluble phenols, etc., whose content is affected by both pre- and post-harvest factors [3,4,5,6].

Carotenoids represent by far the most studied components of tomato fruits [7], given that they are considered as the main dietary source of lycopene [8], one of the two dominant components, along with β-carotene [9], which are responsible for the characteristic color of the ripe fruits. From a nutritional point of view, lycopene is a powerful antioxidant, and its intake has been linked to reduced incidence and severity of several types of cancer and heart disease [10], and β-carotene exhibits strong chemoprotective functions and the highest activity of provitamin A in human metabolism [11].

Color is the most important external trait for assessing the ripening stage and post-harvest life of the fruits and, in turn, affects the decision of the consumer upon its purchase [12]. The color changes from green to orange and then to red that are observed during the ripening of the tomato fruits, are due to the synthesis of carotenoids, specifically lycopene and β-carotene, as well as the breakdown of chlorophyll, due to the conversion of chloroplasts into chromoplasts [13]. The tomato fruit is characterized as climacteric, a feature that enables its harvest at the mature green stage and its ripening even after being detached from the plant. The fruits, which are usually harvested unripe, have a high firmness initially, making them resistant to post-harvest handling but decreases during their ripening [14]. The consumption of the tomato fruits occurs when they have acquired a red color, but without having softened too much yet. This softening is due to the synergistic action of several enzymes on the polysaccharides of the cell walls and results, therefore, in the reduction in the post-harvest life of the fruits [15].

The external color of the tomato fruits is determined by colorimeters that measure the lightness (L *), as well as the a * and b * parameters. According to Shewfelt [16], there is variation in the color, which is perceived by humans, in relation to the one that is determined by the colorimeters. In particular, humans perceive color with complex concepts, such as intensity (hue angle, h^o^) and saturation (chroma, C *). During the ripening of the tomato fruits, the parameter a * and the ratio a */b * increase, while the hue angle, the parameter b * and the lightness (L *) decrease [17,18].

The firmness is one of the most important characteristics of the quality of the tomato fruits and depends on the structure and integrity of the cell walls in the pericarp, as well as on other changes that take place in the cell membranes on the stage of ripening, the harvesting period and the storage duration [6,14,19,20]. During the ripening of the tomato fruits, several changes take place in the structure of the polysaccharides of the cell walls (pectins, hemicelluloses, celluloses), resulting in a decrease in the firmness [21]. These changes are accelerated by the enzymes’ activity, such as of polygalacturonases, hydrolases and lyases [22]. Although the firmness decreases during storage, fruits produced from modern commercial tomato cultivars remain firm for several weeks after harvest [14]. This ability has been attributed to the development of the pericarp, as well as to other changes that occur in the skin of the fruit during storage [23].

Among the various types of tomatoes, the small-fruited (cherry) ones (*Solanum lycopersicum* var. *cerasiforme* (Dunal) DM Spooner, GJ Anderson, RK Jansen) [24] have outperformed the others due to their advanced nutritional value. Cherry tomato varieties are characterized by higher levels of soluble solid and dry matter content than the normal-sized ones due to the negative relationship between fruit size and sugar content [25,26]. These differences are related to the increased content of cherry tomato fruits in sugars (fructose and glucose) and organic acids (citric and malic), which are the most important factors that determine the sweetness, acidity and intensity of flavor [27].

Previous research has shown that the content of all these components in cherry tomato fruits that determine their quality and nutritional value is influenced by genetic (such as cultivar) and environmental factors, as well as by cultivation practices, temperature and light levels in the growing environment, the harvesting period, the grafting on various rootstocks, the irrigation method, the type of substrate, the composition of the nutrient solution, the number of fruits per truss, the size of the fruit, the conditions and duration of the storage, the postharvest treatments [28,29,30,31,32,33,34,35,36,37,38,39,40,41,42,43,44] and others.

However, the reported results are often contradictory, making it impossible to draw solid conclusions about the overall effects of these factors on cherry tomato quality. This is mainly attributed to the existing complex interactions involving genotype, growth environment and storage conditions [36,39]. Moreover, apart from these, a factor that has never been examined in all the above research [28,29,30,31,32,33,34,35,36,37,38,39,40,41,42,43,44] concerns the position of the fruit within the truss, given that in all the published studies the fruits used for the determinations were randomly selected from the harvested ones. Due to the different conditions of exposure to light, as well as to the supply of photosynthetic products, water and nutrients, the fruits located in different positions along the truss are likely to have different quality and nutritional composition.

It is obvious that the harvesting period and storage period significantly affect the quality traits and nutritional components of cherry tomato fruits. However, the effect of the fruit position in the truss, as well as the relative contribution of each one of the above factors, is not known. Thus, the aim of the present study was to investigate the effect (if any) of the fruit position in the truss and its relative contribution, in relation to the harvesting period and duration of storage, on the quality traits and nutritional components of cherry tomato.

## 2. Results

### 2.1. Skin Color Parameters L *, C *, h^o^ and a */b * Ratio

The fruit skin color parameter L * was significantly affected by the harvesting period and the storage duration, and their relative contribution was similar (η^2^ = 19 and 17, respectively) (Table 1). As an average of the three storage durations (0, 4 and 10 days) and the two positions of the fruit in the truss (base and top), the fruits had the lowest value of the color parameter L * (37.8) in the May harvest, which increased significantly in the following June harvest (39.0) but remained unchanged thereafter (39.3). On the other hand, as an average of the three harvesting periods (May, June and July) and the two positions of the fruit in the truss, the fruits had the highest value of the color parameter L * (39.6) at the day of harvest, which decreased significantly after 4 days of storage (38.0) but remained unchanged thereafter (38.5).

The fruit skin color parameter C * was significantly affected only by the harvesting period, while the interaction harvesting period × storage duration was also significant, and their relative contribution was η^2^ = 30 and 15, respectively (Table 1). As an average of the three storage durations and the two fruit positions in the truss, the fruits had the lowest value of the color parameter C * (30.2) in the May harvest, which increased significantly in the following (June) harvest (34.8) but remained at the same levels thereafter (33.2). Regarding the significant interaction harvesting period × storage duration, as an average of the two fruit positions in the truss, only fruits harvested in June exhibited a significant increase in the color parameter C * after 4 and 10 days of storage (38.2 and 35.1, respectively), in comparison to the values on the day of harvest (31.2) (Table 1).

On the other hand, both the fruit skin color parameter h^o^ and the a */b * ratio were significantly affected only by the fruit position in the truss (Table 1). As an average of the three harvesting periods and the three storage durations, the fruits at the base of the truss had the lowest value of the color parameter h^o^ (47.5) and the highest a */b * ratio (0.92) compared to the fruits at the top (50.4 and 0.83, respectively).

### 2.2. Fruit Pigments

Both lycopene and total carotenoids of the fruits were significantly affected by the harvesting period and the storage duration, but the relative contribution of the storage duration was higher than that of the harvesting period (η^2^ = 54 and 21 for lycopene and η^2^ = 55 and 19 for total carotenoids, respectively) (Table 1). As an average of the three harvesting periods and the two fruit positions in the truss, the fruits at the day of harvest had the lowest value of both lycopene (19.0 μg/g f.w.) and total carotenoids (39.9 μg/g f.w.), which increased significantly after 4 (21.5 and 44.2 μg/g f.w., respectively) and 10 days of storage (25.3 and 51.1 μg/g f.w., respectively). On the other hand, as an average of the three storage durations and the two positions of the fruit in the truss, the highest value of both lycopene (24.0 μg/g f.w.) and total carotenoids (48.0 μg/g f.w.) was observed in the fruits harvested in June and the lowest in the ones harvested in May (20.0 and 41.5 μg/g f.w., respectively).

On the other hand, β-carotene of the fruits was only significantly affected by the harvesting period, while the interaction harvesting period × storage duration was also significant, and their relative contribution was η^2^ = 55 and 12, respectively (Table 1). As an average of the three storage durations and the two positions of the fruit in the truss, the fruits had the lowest value of β-carotene when harvested in May (15.8 μg/g f.w.), which increased significantly in the following harvest in June (18.6 μg/g f.w.) but remained unchanged thereafter (18.2 μg/g f.w.).

### 2.3. Fruit Firmness

The firmness of the fruits was significantly affected only by the storage duration (Table 2). As an average of the three harvesting periods and the two fruit positions in the truss, the fruit firmness significantly increased after 10 days of storage (1.76 from 1.59 kg).

### 2.4. Fruit Antioxidant Capacity

The fruit antioxidant capacity was only significantly affected by the harvesting period, while the interaction harvesting period × storage duration was also significant, although the relative contribution of the harvesting period was higher (η^2^ = 77) (Table 2). As an average of the three storage durations and the two fruit positions in the truss, lowest antioxidant capacity (26.1 mg AAE/100g f.w.) was found in the fruits harvested in May, and highest in June and July harvests (44.4 and 52.8 mg AAE/100g f.w., respectively). Regarding the significant interaction harvesting period × storage duration, only the fruits harvested in June exhibited a significant decrease after 4 days of storage (48.4 mg AAE/100g f.w.), comparing to the value on the day of harvest (58.6 mg AAE/100 g f.w.) (Table 2).

### 2.5. Fruit pH and Titratable Acidity

The fruit pH was significantly affected only by the fruit position in the truss, while the interaction harvesting period × storage duration was also significant, the relative contribution of which was equal (η^2^ = 23) (Table 2). As an average of the three harvesting periods and the three storage durations, the fruits at the base of the truss had the highest pH (4.59) compared to the ones at the top (4.53). Regarding the significant interaction harvesting period × storage duration, as an average of the two fruit positions in the truss, only the fruits harvested in June showed a significant decrease after 4 days of storage (4.48) compared to the day of harvest (4.62) (Table 2).

On the other hand, the fruit titratable acidity was significantly affected by both the harvesting period and the fruit position in the truss, as well as by the interaction harvesting period × storage duration (Table 2). The relative contribution of the interaction was higher than that of the harvesting period and the fruit position in the truss (η^2^ = 23, 20 and 15, respectively) (Table 2). As an average of the three storage durations and the two positions of the fruits in the truss, the fruits harvested in May had the lowest acidity (0.098% citric acid), which increased significantly in fruits harvested in June (0.108% citric acid) but remained unchanged thereafter (0.114% citric acid). As an average of the three harvesting periods and the three storage durations, the fruits at the base of the truss had the lowest acidity (0.102% citric acid) compared to the ones at the top (0.112% citric acid). Regarding the significant interaction harvesting period × storage duration, the fruits harvested in June showed a significant decrease from the 4th to the 10th day of storage (from 0.123 to 0.098% citric acid), while contrarily the ones harvested in July showed a significant increase from the 4th to the 10th day of storage (from 0.103 to 0.125% citric acid) (Table 2).

### 2.6. Fruit Dry Matter, Total Soluble Solids and Total Soluble Phenols

All three of these parameters were only significantly affected by the harvesting period, while the interaction harvesting period × storage duration was also significant for dry matter and total soluble phenol content. However, the relative contribution of the harvesting period was higher (η^2^ = 51 and 68, respectively) than that of the interaction (η^2^ = 14 and 9, respectively) (Table 2).

As an average of the three storage durations and the two fruit positions in the truss, the lowest dry matter and total soluble solid content was found in fruits that were harvested in May (8.13 and 7.62%, respectively) and increased significantly in June (9.82 and 9.19%, respectively) but remained unchanged thereafter (9.97 and 9.08%, respectively). Regarding the significant interaction harvesting period × storage duration, only the fruits harvested in June showed a significant decrease in the total soluble solid content after 4 days of storage (8.27%), compared to the levels on the day of harvest (9.60%) (Table 2).

As an average of the three storage durations and the two fruit positions in the truss, the fruits had the lowest total soluble phenol content (0.28 mg GAE/g f.w.) when harvested in May, which increased significantly in the following harvests (0.39 and 0.45 mg GAE/g f.w. in June and July, respectively). Regarding the significant interaction harvesting period × storage duration, as an average of the two fruit positions in the truss, only the fruits that were harvested in June showed a significant decrease in the total soluble solid content after 4 days of storage (8.27%), compared to the value on the day of harvest (9.60%) (Table 2).

## 3. Discussion

The effect of environmental conditions during the harvesting period and storage on the quality characteristics of cherry tomato fruits have already been studied and discussed in detail [28,29,30,31,32,33,34,35,36,37,38,39,40,41,42,43,44]. In brief, it is well known that both temperature and irradiance during the harvesting period affect biosynthesis pathways of primary and secondary metabolites [6,28,43,44,45], and thus final fruit composition, with secondary metabolites having antioxidant properties were the most sensitive [28,44,45]. On the other hand, the cherry tomato fruit is characterized as climacteric, and thus undergoes many physiological and biochemical processes during its postharvest life that affect most of the quality attributes [29]. In this context, storage conditions are critical factors to maintain quality of the cherry tomato fruit in order to slow down the ethylene-driven ripening process [29,38,46].

The present study focuses on the quality evaluation of the cherry tomato fruits in relation to the position of the fruit in the truss and the relative contribution of this factor. When the three factors of harvesting period, storage duration and fruit position in the truss were taken into simultaneous consideration, the ANOVA (Table 1 and Table 2) indicated that the harvesting period had the most significant effect on skin color parameters L * and C * and β-carotene, as well as on antioxidant capacity, total soluble phenols, dry matter and total soluble solids, since most of the variation in the data accounted for this factor while it had an appreciable effect on titratable acidity as well. The storage duration had a dominant effect on firmness, total carotenoids and lycopene while it had an appreciable effect on skin color parameter L * as well. On the other hand, the fruit position in the truss exerted an exclusive effect on skin color parameter h^o^ and a */b * ratio and pH while it also had an appreciable effect on titratable acidity. The most important and significant interaction was exhibited between the harvesting period and the storage duration. Most of the variation in pH and titratable acidity data were accounted for by this interaction.

The increased fruit dry matter, as well as total soluble solid and total soluble phenol (both constituents of dry matter) content observed in the June and July harvests compared to those in the May harvest is attributed to the higher temperatures and sunshine that prevail during these periods. Similar results in cherry tomato have been reported by other researchers [28,44]. It is hypothesized that higher temperatures, which enhance transpiration and reduce fruit water content and indirectly increase dry matter content [44], as well as total soluble solids and total soluble phenols, all of which were expressed on fresh weight basis. Seasonal variation was also observed in fruit antioxidant capacity, as well as in lycopene, total carotenoid and β-carotene (all three components that contribute to fruit antioxidant capacity [10,11]) content, with fruits harvested in June and July having a higher content compared to those harvested in May. This is probably due to the more favorable environmental conditions (higher temperatures and irradiation) for the biosynthesis of these secondary metabolites [28,43,44], and also to the increased availability of components (carbohydrates) necessary for their biosynthesis [28]. However, it has been reported that the effect of environmental conditions on the secondary metabolites of cherry tomato fruits is particularly complex compared to that on the primary metabolites, as it is also related to plant management and fruit load, and thus the sink-source ratio [28].

It has been reported that a long storage period reduces fruit firmness of both the normal-sized [14] and the small-fruited (cherry) [38] tomato fruits, but this fact was not observed in the present study. This is probably a result of storing at the appropriate temperature (12 °C) for a limited duration (10 d) and also of the characteristics of the variety used, given that a strong effect of all these factors, as well as of the storage temperature x genotype interaction on the firmness of the small-fruited tomato, has been reported [38]. In addition, it has been suggested [38] that the ability of small-fruited cultivars to maintain the firmness during their post-harvest life, apart from the special characteristics (ratio between the volume of the fruit and its external transpiration surface), depends on other factors and that multiple factors are involved in the processes related to the disassembly of polysaccharides in the primary cell wall, making fruit softening a complex process.

The vivid red color of the skin of the tomato fruits is perceived as the main appearance characteristic that is related to the ripening stage and is also associated with the quality, given that it is a basic requirement of the quality standards of the European Union and influences consumer preference during the purchase of the product [46]. The colorimeters are used for its measurement providing specific values (L *, a * and b *), although the a */b * ratio has been reported to correlate better with the human perception of color [47,48]. Indeed, according to the USDA-based color classification [49], the a */b * values in the range of 0.60–0.95 indicate the recommended color at the marketable stage of the fruit [47].

In the present study, fruit sorting was performed based on color, visually perceived by the human eye. However, it was concluded that differences in visual color are difficult to be perceived with the human eye, given that in fruits visually classified as the same color, deviations in the a */b * ratio were found after colorimeter measurement, and indeed they were only significant between fruit positions in the truss (Table 1), with the ones at the base having a higher value compared to those at the top. Similar differences were also reflected in the color h^o^ parameter (Table 1).

However, no corresponding differences were observed in the lycopene content of fruits as it was not significantly affected by the position of the fruit in the truss (Table 1), despite the fact that the ratio a */b * has been well correlated with lycopene content [17,50,51]. This is apparently due to the fact that the color measurement is performed on the surface of the skin fruit, while the determination of lycopene is on the whole fruit and, furthermore, it is a given that the ratio of the skin to the flesh is very low.

The pH and titratable acidity are interrelated parameters, yet each of them is analytically determined in separate ways, and they additionally provide their own information about tomato fruit quality, with titratable acidity being an indicator of acid content [52] and, therefore, taste, especially relatively with sugars [49]. The pH is the negative log (base 10) of the hydrogen cations concentration (H^+^) in the juice of the fruit that dissociates from acids, while titratable acidity measures total acidity (sum of H^+^ and undissociated acids) by titrating them with a standard base (usually NaOH) and is expressed as the per cent of the predominant organic acid [52].

The titratable acidity of tomato fruits is shaped by organic acids (citric, malic and glutamic acid being the most common ones) [52,53,54,55], which are produced in the Krebs cycle (along with their derivatives, fatty acids and amino acids) [52]. However, inorganic acids, such as phosphate, often play an important role as well [50,54]. When the acidity of tomato juice was calculated from the individually measured concentrations of citric, glutamic, malic and phosphoric acids, the titrated values were equal to 86% of the measured one, indicating that 14% of the titratable acidity is attributed to other acids that were not included in the above ones (e.g., ascorbate and oxalate, as well as numerous amino acids, including aspartic and γ-aminobutyric). In some tomato cultivars, these last two amino acids have been shown to be present at levels comparable to those of glutamic acid in ripe fruit [55]. Malic acid is typically present at only one-tenth of the level of citric acid, although the ratio of malic to citrate can vary significantly between different tomato varieties [50,53,54]. Citric is, therefore, clearly the dominant acid in tomato fruit [44,50] and the largest contributor to titratable acidity, with typical content in large tomato fruit in the range of 0.2–0.6% [52].

Previous studies have already examined the effects of cultivar, fruit ripening and fruit storage on pH and titratable acidity [50,53,56,57,58,59,60,61,62,63]. Tomato fruits are considered as low pH fruits (<4.6) [64].

It has been indicated that pH increases as the fruit ripens while titratable acidity simultaneously decreases [50,53,57,59,61,62,63] and this reduction is mainly due to a degradation of citric acid in the fruits [50], given that no changes during ripening have been reported in glutamic and phosphoric acid [50,54]. During tomato fruit ripening, part of this change may be due to the metabolic conversion of acids to sugars through gluconeogenesis [65] or entirely to the respiration process [50], given that a loss of titratable acidity has been positively correlated with a higher respiration rate when organic acids are used as a substrate [60].

In summary, although some cultivation practices (such as transplanting or direct seeding, drip or furrow irrigation, organic or conventional cultivation) [50,66], environmental factors [67] and the ripening process [50,51,61,62,63] decisively affect fruit pH, the higher metabolic rate of the basal fruits could be the reason for the lower titratable acidity and increased pH values compared to the top fruits observed in the present study.

All the determined parameters were compared with the corresponding range of values reported in the literature for each one [28,29,31,32,33,34,35,36,37,38,39,40,41,43,44], and, as expected, exclusions were observed, given that these parameters are influenced by several factors, but also the interactions between them, as already mentioned at length in the introduction and in the discussion sections.

## 4. Materials and Methods

### 4.1. Experimental Design and Fruit Sampling

The experiment was conducted at Agris S.A., in Northern Greece from 1st February–30th July. Cherry tomato Genio F1 plants grafted onto Defensor rootstock were cultivated hydroponically in rockwool slabs in a heated glasshouse. Throughout the cultivation, the temperature in the glasshouse was maintained in the range of 15–28 °C (17.6 ± 1.76 °C minimum and 27.1 ± 1.82 °C maximum), by activating the heating or shading system or opening the top roof windows. Bumblebees (*Bombus terrestris* L.) were introduced for fruit setting (20 hives/ha) while trusses were trimmed so that each one supported only 10 fruits.

Whole trusses at the red ripe stage of ripeness were harvested at three harvest times 27 May, 14 June and 1 July, transferred to the facilities of the Lab of Vegetable Crops of the Aristotle University of Thessaloniki and were stored at 12 °C and 65–75% R.H. On the day of harvest (day 0) and after 4 and 10 days of storage, the first four (1st-4th) and the last four (7th–10th) fruits of each truss were grouped and the external color and the firmness of each fruit were measured; fruits were then analyzed for the determination of antioxidant capacity, pH and titratable acidity, as well dry matter, total soluble solid, total soluble phenol, lycopene, total carotenoid and β-carotene content. In each treatment, three trusses (replicates) were used.

### 4.2. Measurement of Fruit Skin Color

Skin color was measured at two diametrically opposite spots at the equator of the fruit according to Mitsanis et al., 2021 [68]. Color changes were quantified in the L *, a * and b * color space. Hue angle (h = 180 + tan^−1^ (b */a *) and chroma values (C * = (a *^2^ + b *^2^)^1/2^) were calculated from a* and b* values. L* refers to lightness, ranging from 0 = black to 100 = white; hue angle (h^o^) value is defined as a color wheel, with red-purple color at an angle of 0°, yellow color at 90°, bluish-green color at 180° and blue color at 270°, and chroma (C *) represents color saturation, which varies from dull (low values) to vivid (high values) [69].

### 4.3. Measurement of Firmness

Fruit firmness was measured at two diametrically opposite spots at the equator of the fruit according to Mitsanis et al., 2021 [68].

### 4.4. Sample Preparation and Determinations

Sample preparation and determinations were performed, as described by Mitsanis et al., 2021 [68].

#### 4.4.1. Dry Matter, pH, Titratable Acidity, and Total Soluble Solids

Determinations were performed, as described by Mitsanis et al., 2021 [68]. The titratable acidity was expressed as % of citric acid, given that the major organic acid was citric acid [44,50,70].

#### 4.4.2. Carotenoids

Lycopene, β-carotene and total carotenoid content were determined using the methods of Lichtenhaler and Wellburn [71] and Fish [72] and described in detail by Mitsanis et al., 2021 [68].

For the individual determination of the pigments, the following equations were used:Lycopene (μg/g) = (3.521 × Abs503 − 0.587 × Abs450) × V/W (1)
β-carotene (μg/g) = (4.367 × Abs450 − 2.947 × Abs503) × V/W (2)
Total carotenoids (μg/g) = (1000 × Abs470 × V/W) − ((2.27 × Chl a) − (81.4 × Chl b)) × 227 (3)
where Abs = absorbance, V = extract volume and W = weight of homogenized tissue.

#### 4.4.3. Total Soluble Phenols and Antioxidant Capacity

For the determination of total soluble phenolic compounds and antioxidant capacity, 5 g of homogenized tissue were mixed with 25 mL 80% methanol and filtered through a Whatman No. 1 filter. Total soluble phenols were determined photometrically according to the method of Scalbert et al. [73], using a standard curve of gallic acid (y = 211.573x − 2.604, r = 0.9859) and total antioxidant capacity was determined according to the method of Brand-Williams et al. [74], using a standard curve of ascorbic acid (y = 0.115x + 0.001, r = 0.9994). Both determinations are described in detail by Mitsanis et al., 2021 [68].

For the determination of the total soluble phenols and the antioxidant capacity, the following equations were used:TSP (mg/g) = −7.732 + 214.81 Abs760 × V × 100/W (4)
AC (mg/100 g) = 0.001 + 0.115 ΔAbs518 × V × 1000/W (5)
where Abs = absorbance, V = extract volume and W = weight of homogenized tissue; ΔAbs = difference in the absorbance between the sample and the 0 mg/mL ascorbic acid solution.

### 4.5. Statistical Analyses

A complete randomized factorial design (3 harvesting periods × 3 storage durations × 2 fruit positions in the truss) was used, with three replicates per treatment. Data were subjected to analysis of variance (ANOVA) using the statistical software SPSS v25. The effect size of each factor was evaluated using η^2^ (eta squared) criterion calculated as follows: η^2^ = SS factor/SS total, where SS = sum of squares. The means were separated with the Duncan’s new multiple range test (*p* < 0.05).

## 5. Conclusions

The results of the present study indicated that quality traits and nutritional components of cherry tomato fruits are dependent on several factors, including harvesting period, storage duration and fruit position in the truss, although the relative contribution of each factor varied according to the component. The harvesting period had the most significant effect on skin color parameters L * and C * and β-carotene, as well as on antioxidant capacity, total soluble phenols, dry matter and total soluble solids, while it had an appreciable effect on titratable acidity. The storage duration had a dominant effect on firmness, total carotenoids and lycopene, while it had an appreciable effect on skin color parameters L * as well. On the other hand, the fruit position in the truss exerted an exclusive effect on the skin color parameter h^o^, a */b * ratio and pH while it had an appreciable effect on titratable acidity as well. The differences in the color parameter h^o^ and the a */b * ratio that were significant only between fruit positions in the truss, which were, however, not visually perceived by the human eye, indicate a different physiological maturity and, therefore, a different metabolic activity and ripening stage, which is probably also the reason for the differences in the pH and the titratable acidity of the fruits in the two positions of the truss. That still needs to be clarified in the future.

## Figures and Tables

**Table 1 plants-12-00315-t001:** ANOVA for color skin parameters lightness (L *), chroma (C *), hue angle (h^o^) and a */b * ratio, as well as lycopene (Lyc, μg/g f.w.), total carotenoids (T-car, μg/g f.w.) and β-carotene (β-car, μg/g f.w.), of cherry tomato fruits harvested in three dates during the period of May–July and stored at 12 °C for 0, 4 and 10 days. The fruits were collected from two positions in the truss, the first four fruits (base) and the last four ones (top).

			L *		C *		h^o^		a */b *		Lyc		T-Car		β-Car	
Source of Variance		DF	*P*	η^2^	*P*	η^2^	*P*	η^2^	*P*	η^2^	*P*	η^2^	*P*	η^2^	*P*	η^2^
Harvesting period (A)		2	*	19	***	30		5		5	***	21	***	19	***	55
Storage duration (B)		2	*	17		6		2		2	***	54	***	55		0
Fruit position (C)		1		0		4	***	39	***	40		3		4		2
A × B		4		14	*	15		10		10		2		3	*	12
A × C		2		2		2		1		1		0		0		0
B × C		2		1		1		0		0		0		0		2
A × B × C		4		2		2		7		7		0		0		0
Error		36														
Harvesting period	May June July		37.8 39.0 39.3	b a a	30.2 34.8 33.2	b a a	48.6 48.5 49.8		0.89 0.89 0.85		20.0 24.0 21.8	c a b	41.5 48.0 45.7	b a a	15.8 18.6 18.2	b a a
Storage duration	0 4 10		39.6 38.0 38.5	a b b	31.6 33.5 33.2		49.5 48.4 49.0		0.86 0.89 0.87		19.0 21.5 25.3	c b a	39.9 44.2 51.1	c b a	17.6 17.4 17.7	
Fruit position	Base Top		38.7 38.7		33.4 32.1		47.5 50.4	b a	0.92 0.83	a b	22.4 21.4		46.1 44.0		17.7 17.4	
May	0 4 10		38.3 38.0 37.1	bcd cd d	30.1 29.5 31.1	cd d cd	48.8 48.3 48.6	ab ab ab	0.88 0.89 0.88	ab ab ab	18.0 19.2 22.9	f ef bcd	38.0 40.5 45.9	c c b	16.0 15.8 15.6	c c c
June	0 4 10		39.7 38.7 38.6	abc bcd bcd	31.2 38.2 35.1	cd a ab	50.1 46.2 49.1	a b a	0.84 0.96 0.87	b a b	20.3 23.6 28.1	def bc a	41.9 46.4 55.8	bc b a	18.0 19.4 18.4	ab a ab
July	0 4 10		40.7 37.4 39.9	a d ab	33.5 32.7 33.4	bc bcd bc	49.5 50.8 49.3	a a a	0.86 0.82 0.87	b b b	18.8 21.1 24.9	f cde b	39.7 45.7 51.6	c b a	18.6 16.9 19.0	a bc a

DF, degrees of freedom; *P*, probability; η^2^, eta squared; * significant effect at the 0.05 level; *** significant effect at the 0.001 level. Different letters following values within each column indicate significantly different values at 0.05 level according to the Duncan’s multiple range test.

**Table 2 plants-12-00315-t002:** ANOVA for firmness (F, kg), antioxidant capacity (AC, mg ascorbic acid equivalents/100 g f.w.), pH, titratable acidy (TA, % citric acid), dry matter (DM, %), total soluble solids (TSS, %) and total soluble phenols (TSP, mg gallic acid equivalents/g f.w.) of cherry tomato fruits harvested in three dates during the period of May–July and stored at 12 °C for 0, 4 and 10 days. The fruits were collected from two positions in the truss, the first four fruits (base) and the last four ones (top).

			F		AC		pH		TA		DM		TSS		TSP	
Source of Variance		DF	*P*	η^2^	*P*	η^2^	*P*	η^2^	*P*	η^2^	*P*	η^2^	*P*	η^2^	*P*	η^2^
Harvesting period (A)		2		11	***	77		7	**	20	***	51	***	49	***	68
Storage duration (B)		2	**	24		0		2		0		2		3		2
Fruit position (C)		1		2		2	**	23	*	15		1		6		2
A × B		4		11	*	6	**	23	**	23	*	14		8	*	9
A × C		2		0		0		3		2		0		2		0
B × C		2		1		0		1		0		1		0		1
A × B × C		4		5		0		2		1		0		1		1
Error		36														
Harvesting period	May June July		1.57 1.67 1.70	b ab a	26.1 44.4 52.8	c b a	4.59 4.56 4.54		0.098 0.108 0.114	b a a	8.13 9.82 9.97	b a a	7.62 9.19 9.08	b a a	0.28 0.39 0.45	c b a
Storage duration	0 4 10		1.59 1.58 1.76	b b a	42.1 41.0 40.2		4.58 4.55 4.56		0.106 0.107 0.107		9.46 9.09 9.40		8.77 8.39 8.72		0.38 0.36 0.38	
Fruit position	Base Top		1.62 1.67		42.6 39.7		4.59 4.53	a b	0.102 0.112	b a	9.41 9.23		8.84 8.42		0.39 0.36	
May	0 4 10		1.58 1.45 1.67	bc c b	22.4 26.8 29.2	d d d	4.59 4.60 4.57	abc ab abc	0.100 0.095 0.098	cd d d	8.09 8.03 8.28	d d d	7.55 7.55 7.75	c c c	0.28 0.27 0.28	c c c
June	0 4 10		1.58 1.72 1.73	bc ab ab	45.4 47.8 40.0	bc bc c	4.62 4.48 4.58	a d abc	0.100 0.123 0.098	cd ab d	9.58 10.37 9.52	cd bc bc	9.15 9.37 9.05	ab a ab	0.37 0.42 0.38	b b b
July	0 4 10		1.62 1.58 1.90	bc bc a	58.6 48.4 51.5	a b ab	4.51 4.58 4.53	cd abc bcd	0.118 0.103 0.125	abc bcd a	10.71 8.86 10.41	ab ab a	9.60 8.27 9.37	a bc a	0.50 0.38 0.48	a b a

DF, degrees of freedom; *P*, probability; η^2^, eta squared; * significant effect at the 0.05 level; ** significant effect at the 0.01 level; *** significant effect at the 0.001 level. Different letters following values within each column indicate significantly different values at 0.05 level according to the Duncan’s multiple range test.

## Data Availability

Not applicable.

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
