# Peer review of "Quality Traits and Nutritional Components of Cherry Tomato in Relation to the Harvesting Period, Storage Duration and Fruit Position in the Truss"

_plants, 2023, doi:10.3390/plants12020315_

Round 1

Reviewer 1 Report

This manuscript analyzed the effects of the harvesting period, storage duration and the fruit position on the truss on some quality and nutritional parameters of cherry tomato. The results showed that such parameters were less affected by the position of the fruits in the truss. This manuscript is very well written and structured and fits very well within the scope of the Special Issue “Pre and Postharvest Physiology and Biochemistry of Fresh Fruits and Vegetables”. I also consider that the manuscript is of interest to future readers, but I do have serious concerns about the interpretation of Results and the Discussion section. The authors should focus on the interactive effects of all factors instead of analysing them individually. Moreover, they should discuss their results in relation to published literature instead of including some broad generalizations that are not based upon the results presented. For example, the authors mentioned that “the harvesting period and the storage duration have a significant effect on the quality characteristics of tomato fruits”, but this fact was not observed in all tested parameters and was not discussed. The authors also mentioned that long storage period reduces fruit firmness, but this fact was not observed in the present study and it was not discussed. Moreover, there are other parameters (F, AC, DM, TSS and TSP) that were not mentioned in the Discussion. The implications of these results for the tomato cultivation and production should also be discussed. Because of these, I recommend a substantial revision of the Discussion section.

Reviewer 2 Report

Please, provide an adequate reference for the equation used to calculate lycopene. All my suggestions that could be useful for improving the manuscript are given in the comments.

Reviewer 3 Report

In the storage conditions of vegetables, carbon dioxide and moisture content should be given in addition to the ambient temperature. How was it stored in what containers? What percentage of deterioration occurred in the storage strength from properties such as decay and deterioration. Information on these issues should be given.

Reviewer 4 Report

The manuscript presents an interesting idea of “Quality Traits and Nutritional Components of Cherry Tomato in Relation to the Harvesting Period, Storage Duration and Fruit Position in the Truss”. Unfortunately the paper contains many approaches that need to be reviewed in such a manner that the paper to gain more precision and clarity.

Results

Please mention the unit of measure for all the parameters from Table 1 and Table 2. In addition to the measured values, it is also necessary to indicate their standard deviations.  What harvest period is for the results presented in Table 1 and Table 2, for the storage duration and fruit position (May/June/July)? Correlate and explain the obtained results.

Please explain the results obtained during storage period: for ex. total soluble phenols with contents lower than 0.50 mg GAE per g of fresh weight; β-carotene content.  The detection limit for the instability for each studied parameter must be presented.

The equation of the calibration curve of gallic and ascorbic acid must be included.

 You have to clarify why you have chosen these methods for pigments determination and correlate all the results obtained with literature data.

Materials and methods

Please mention what are the analytical standards used for the determinations.

   For individual determination of the pigments (equations 1,2,3) please explain each parameter what represents. Please also indicate the references, since I could not identify these formulas in the indicated ones (71, 72). The unit of measure for volume and weight of homogenized tissue must be specified.

3.      Conclusion

The authors have to explain more precise their contributions to knowledge and if it is possible to underline the scientific aspects that still need to be clarified in the future.  
